# Drug-Checking and Monitoring New Psychoactive Substances: Identification of the U-48800 Synthetic Opioid Using Mass Spectrometry, Nuclear Magnetic Resonance Spectroscopy, and Bioinformatic Tools

**DOI:** 10.3390/ijms26052219

**Published:** 2025-02-28

**Authors:** Maria Beatriz Pereira, Carlos Família, Daniel Martins, Mar Cunha, Mário Dias, Nuno R. Neng, Helena Gaspar, Alexandre Quintas

**Affiliations:** 1Centro de Investigação Interdisciplinar Egas Moniz, Cooperativa de Ensino Egas Moniz, 2825-084 Caparica, Portugal; mbpereira.lcfpem@egasmoniz.edu.pt (M.B.P.); carlosfamilia@egasmoniz.edu.pt (C.F.); jmdias@egasmoniz.edu.pt (M.D.); nneng@egasmoniz.edu.pt (N.R.N.); 2Laboratório de Ciências Forenses e Psicológicas Egas Moniz, Campus Universitário—Quinta da Granja, Monte de Caparica, 2825-084 Caparica, Portugal; 3Centro de Química Estrutural, Institute of Molecular Sciences, Departamento de Química e Bioquímica, Faculdade de Ciências, Universidade de Lisboa, Campo Grande, 1749-016 Lisboa, Portugal; 4Kosmicare, Rua de São Dionísio, 17, 4000-027 Porto, Portugal; daniel.martins@kosmicare.org (D.M.); mar.cunha@kosmicare.org (M.C.); 5BioISI—Biosystems & Integrative Sciences Institute, Departamento de Química e Bioquímica, Faculdade de Ciências, Universidade de Lisboa, Campo Grande, 1749-016 Lisboa, Portugal; hmgaspar@ciencias.ulisboa.pt

**Keywords:** U-type opioids, U-48800, U-51754, gas chromatography coupled with mass spectrometry, nuclear magnetic resonance, molecular dynamics

## Abstract

The misuse of opioids and opiates has remained a persistent issue since the 19th century. The recent resurgence of non-fentanyl synthetic opioids, such as U-type opioids and nitazenes, has further exacerbated the ongoing crisis. Identifying these synthetic opioids presents many challenges, including the emergence of new substances, the lack of standards, and the presence of structural isomers. This highlights the need for a robust structural characterisation strategy in forensic laboratories. To address these challenges, we developed a methodology to identify a U-type opioid sample received by Kosmicare from the European Union-funded SCANNER project, which was suspected to be either U-48800 or U-51754. Our innovative approach combined gas chromatography coupled with mass spectrometry (GC-MS), nuclear magnetic resonance spectroscopy (NMR), and molecular dynamics to characterise the questioned sample unequivocally. While the GC-MS analysis suggested a potential match with the mass spectrum of U-51754 and its structural isomer U-48800, NMR analysis confirmed the presence of U-48800 in the sample, which was further validated through molecular dynamics experiments. These experiments provided additional insights, confirming the structural features underlying the obtained NMR profile. The presented methodology offers a valuable solution for cases involving the identification of isomers, which are currently one of the most significant challenges in identifying new psychoactive substances.

## 1. Introduction

The misuse of synthetic psychoactive drugs remains one of the most critical contemporary global issues, posing significant challenges to international drug policies [1]. The recent surge in synthetic opioids in illicit markets is closely tied to the opioid crisis that emerged in North America in the late 1990s. Central to this crisis was oxycodone, a semi-synthetic derivative of morphine, which was improperly marketed by Purdue Pharmaceuticals LP and Xenodyne Pharmaceutical Inc. as a non-addictive analgesic. This misleading marketing led to a sharp increase in opioid prescriptions, thereby fuelling widespread addiction [2,3]. As a result, the heroin market saw a resurgence in 2010 to meet the demand of individuals dependent on prescription opioids. The subsequent heroin shortage further fuelled the rise of illicit synthetic opioids, notably fentanyl and its analogues. The increasing potency of new synthetic opioids contributed to a devastating overdose crisis in North America and Canada. Drug-Checking Services (DCS)—where individuals can submit drug samples for chemical analysis—have played a vital role in harm reduction and drug monitoring [4]. This contemporary opioid societal crisis has led to tight regulatory measures around fentanyl and its derivatives. Consequently, non-fentanyl opioid analogues have begun to emerge in illicit drug markets across North America, Europe, Asia, and Australia [5]. Among these substances, U-type opioids (utopioids) have attracted particular attention due to their potential risks [6]. U-type opioids are characterised by the presence of a 1,2-*trans*-cyclohexane-diamino moiety connected to a phenyl or benzyl ring through a carbonyl group (Figure 1) [6].

To date, several U-type opioids have already been identified in illicit drug markets, including U-47700, Isopropyl-U-47700, 4-(trifluoromethyl) U-47700, 3,4-methylenedioxy U-47700, *N*-methyl U-47931E, U-47931E, U-48800, U-49900, U-50488, U-51754, and U-77891 (Figure 2) [5,6]. Despite intense international monitoring, only U-47700 is classified as a Schedule I substance under the 1961 Single Convention on Narcotic Drugs [7]. The deliberate modification of controlled psychoactive substances to produce isomeric variants is a common strategy within the New Psychoactive Substance (NPS) market, enabling circumvention of existing legal frameworks. This practice has been observed previously with synthetic cathinones, where individuals exposed to 3-methylmethcathinone (3-MMC) generated false positive results for mephedrone (4-MMC) during drug screenings [8]. Thus, the emergence of new structural isomers of controlled U-type opioids raises concerns about potential false positive identifications, emphasising the need for rigorous structural characterisation methods in forensic laboratories to distinguish these compounds accurately.

The European Network of Forensic Science Institutes (ENFSI) emphasises the importance of reference materials for accurately identifying psychoactive substances [9]. However, the rapid proliferation of NPS has far outpaced the availability of certified reference standards, complicating the analytical efforts of forensic laboratories. Therefore, alternative methods for substance identification that do not rely on authentic reference standards have become increasingly necessary. Such strategies include comparing substances with spectral reference databases and interpreting data from non-traditional methods, such as nuclear magnetic resonance spectroscopy (NMR) [10]. Within the framework of the European Union-funded SCANNER Project [11], Kosmicare (www.kosmicare.org), a Portuguese non-profit harm reduction organisation, receives several samples of new psychoactive substances, including samples to evaluate the analytical competence of its service. The sample included in this study was subject to an interlaboratory assay, the results of which were not consistent, with some laboratories suspecting it to be U-48800 and others U-51754. Since there are few examples in the literature of U-type opioids’ structural characterisation [12,13,14,15,16,17,18], and some of the existing reports have not undergone peer review [12,13], in this study, we present a comprehensive analytical approach that integrates mass spectrometry (MS), NMR, and molecular dynamics simulations (MDS) to identify the sample suspected to be either U-48800 or U-51754, thereby contributing to the body of knowledge on the structural elucidation of these emerging opioids and providing a standard workflow for future identification efforts.

## 2. Results

### 2.1. GC-MS Analysis

The sample was analysed by GC-MS, and two signals (retention times 5.55 and 13.42 min) were observed. However, due to its abundance and mass spectra, only the signal at 13.42 min was considered when analysing the mass spectrum. As expected, two matches were found when the mass spectrum was compared with the Chemical C. Cayman Spectral Library (v21022022) and the SWGdrug database (v3.9), corresponding to U-51754 or its structural isomer U-48800, both isomeric compounds with a molecular formula of C_17_H_24_C_l2_N_2_O (Figure 2).

The first piece of information was obtained by analysing the mass spectrum in Figure 3A, namely, the molecule’s molecular weight, i.e., *m*/*z* 342, and its base peak at *m*/*z* 84 (C_5_H_10_N^+^). In accordance with Feeney et al., the fragmentation pathway for the *m*/*z* 84 ion has not been identified [19]. However, one potential structure suggests fragmentation across the cyclohexane ring while preserving the amide group.

Nevertheless, the most important set of peaks is at *m*/*z* 159, 161, and 163, which correspond to the characteristic isotopic pattern indicating the presence of two chlorine atoms in the structure, with an abundance ratio of 9:6:1. Figure 3B shows the possible fragment at *m/z* 159 and its transformation to the fragment at *m*/*z* 133. Moreover, peaks at *m*/*z* 110 and 112 suggest the loss of a chlorine atom from the phenyl ring fragment, with an isotopic pattern of 3:1 (abundance 1500:500), which is characteristic of the presence of a single chlorine atom. Additional fragment ions were observed at *m*/*z* 125, 97, 71, and 58, which match those in the spectral library. The *m*/*z* 125 and 97 correspond to the radical cation *N*,*N*-dimethylcyclohexanamine, followed by the elimination of the ethylene group. The *m*/*z* 71 and 58 fragments are suggested to result from the cleavage of the cyclohexane ring.

### 2.2. NMR Structural Characterisation

Despite all the information provided by GC-MS, it did not allow for the unequivocal characterisation of the molecule. NMR spectroscopy was used to identify and confirm which molecule was present in the sample. The rationale underlying this approach was to start assigning ^1^H and ^13^C NMR spectra, followed by 2D NMR spectroscopy of [^1^H-^1^H]-COSY, [^1^H-^1^H]-TOCSY, [^1^H-^1^H]-NOESY, [^1^H-^13^C]-HSQC, and [^13^C-^1^H]-HMBC. The final 3D structure was achieved using molecular dynamics. 

#### NMR Analysis

The presence of a 1,2,4-trisubstituted benzene moiety, which is compatible with the structures of both U-48800 and U-51754 (Figure 4), was first confirmed by the presence of an ABX spin system of three aromatic signals in the ^1^H NMR spectrum (Figure 5), namely, one doublet at 7.49 ppm (d, 1H, *J*_AX_ = 1.9 Hz), one doublet at 7.37 ppm (d, 1H, *J*_AB_ = 8.2 Hz), and one doublet of doublets at 7.32 ppm (dd, 1H, *J*_AB_ = 8.3 Hz and *J*_BX_ = 1.9 Hz). The corresponding sp^2^ carbon ^13^C NMR signals appear at 129.9, 134.2, and 128.3 ppm, respectively (Figure 6 and Figure 7).

Although both compounds have an ABX spin system, the HMBC correlation of the H_A_ signal, one doublet at 7.37 ppm (*J* = 8.12 Hz), with the methylene carbon at 39.5 ppm is only compatible with the structure of U-48800 (Figure 8). This allows the NMR aromatic methine signals at ^1^H/^13^C 7.49/129.9, 7.32/128.3, and 7.37/134.2 ppm to be assigned to positions C-5, C-7, and C-8, respectively, while the signal at 39.5 ppm is assigned to the methylene group at C-2 (Table 1). In the structure of U-51754, the proton H_A_ and C-2 are separated by four bonds, so it would not be possible to see an HMBC correlation between them (Figure 8B).

Notwithstanding the NMR identification, the full NMR assignment of U-48800 presented in Table 1 was made for future reference. Since there is no previous assignment in the literature for this molecule or for other similar U-type opioids, such as U-47700 and U-49000, NMR assignment attempts are incomplete [14,16,17].

The ^13^C APT signal at 39.5 ppm, attributed to the C-2 carbon linked to the aromatic moiety, correlates in the HSQC with the signals from two diastereotopic protons that form an apparent quartet at 3.99 ppm (Figure 9A,B). Its complex hyperfine structure corresponds to an AB spin system of a methylene group, with a geminal coupling constant of 16.8 Hz. In the HMBC spectrum, these protons show cross-peaks with the deshielded carbon signal at 174.2 ppm, which is characteristic of a carbonyl carbon (Figure 9C). This confirms the presence of a methylene group between the aromatic moiety and the carbonyl group and allows the signal at 174.2 ppm to be assigned to the carbonyl carbon at C-1.

The remaining aromatic carbons of the 1,2,4-trisubstituted benzene appear at 136.6, 134.6, and 134.0 ppm. The HMBC experiment allowed for the assignment of the ^13^C signal at 136.6 ppm to C-4 since this signal shows correlations with the aromatic protons at 7.49 (H-5) and 7.37 (H-8) ppm, as well as with the C-2 methylene protons at 3.94 and 4.03 ppm, together with the lack of a correlation with the H-7 proton at 7.32 ppm (Figure 10). However, the overlap of the cross-peaks from the correlations of the carbons at 134.0, 134.2, and 134.6 ppm with the aromatic protons suggests that each of the quaternary signals at 134.0 and 134.6 ppm can be assigned to C-3 or C-6.

Additionally, in the ^1^H NMR spectrum (Figure 5), it is possible to identify three signals corresponding to three *N*-methyl groups at 2.74, 2.94, and 3.09 ppm, which on the HSQC spectrum (Figure 11A), show cross-peaks with the carbons at 37.8, 43.1, and 30.9 ppm, respectively. The HMBC correlations of the proton at 3.10 ppm with the carbonyl at 174.2 ppm and with the methine carbon at 54.2 ppm (Figure 11B, marked in green and purple) allow for the assignment of this ^1^H signal to the methyl group at position C-9′ and the ^13^C signal at 54.1 ppm to the methine at C-1′, which has the corresponding H-1′ at 4.70 ppm (Figure 7). On the other hand, the HMBC correlations of the two other *N*-methyl groups, which are marked in red and blue in Figure 11B, indicate that they are geminal to a nitrogen atom, which is bonded to the quaternary carbon at 65.5 ppm. Thus, these geminal methyl groups are indistinctively assigned to C-7′ and C-8′.

The remaining NMR ^1^H signals (Figure 5) were easily assigned to the cyclohexane moiety through the TOCSY correlations of H-1′ shown in Figure 12A_,_ along with the corresponding carbon ^13^C signals identified by HSQC cross-peaks (Figure 7 and Figure 12B). Despite the overlapping signals of some protons, the presence of three pairs of diastereotopic protons corresponding to three methylene groups can be observed (^13^C/^1^H 25.5/1.86 and 1.42 ppm; 25.0/1.95 and 1.42 ppm; 24.1/2.19 and 1.61 ppm). There is also one pair with quasi-equivalent protons (^13^C/^1^H 30.1/1.79 ppm). This was expected since axial and equatorial protons of substituted cyclohexane have different chemical shifts. Furthermore, apart from the methine group at C-1′ (^13^C/^1^H 54.2/4.70 ppm), the methine group (^13^C/^1^H 65.5/3.65 ppm) linked to the -*N*(CH_3_)_2_ group also belongs to the cyclohexane ring. Both methine proton signals appear as broad multiplets without hyperfine structure, indicating a chemical environment that induces rapid proton spin relaxation.

COSY and NOESY experiments were performed to assign the cyclohexane proton signals. The key correlations observed in the COSY spectrum (Figure 13A) include those between the protons: 4.70 ⇔ 1.79/3.65 ppm; 3.65 ⇔ 2.19/1.61 ppm and 1.61 ⇔ 1.42/1.95 ppm. Key NOESY correlations (Figure 13B) include the correlation between three axial protons on each side of the cyclohexane ring: 4.70 ⇔ 1.61 ⇔ 1.42 and 3.65 ⇔ 1.78 ⇔ 1.42. These correlations allowed for the identification of the axial proton signals of the cyclohexane ring and, consequently, the equatorial protons. These findings, together with the key COSY correlations and the HSQC experiment, unambiguously assigned all the NMR signals of the cyclohexane moiety, as summarised in Table 1.

### 2.3. Molecular Dynamics Simulation

Molecular dynamics simulations provide a powerful tool for exploring the conformational flexibility of molecules. When combined with experimental structural determination methods, molecular dynamics simulations enable us to accurately identify the corresponding molecular 3D structure [20]. In this study, molecular dynamics simulations were used to evaluate the rotation of the aromatic ring around the C-2-C-3 bond and the range of conformations adopted by the non-aromatic ring, using methanol as the solvent to match the NMR experimental conditions.

For the aromatic ring, the simulation revealed two distinct clusters of structures based on the C-1-C-2-C-3-C-4 dihedral angles: one group with negative angles and another with positive angles. Figure 14 illustrates the superimposed structures from each cluster. The first cluster (Figure 14A) contains 5158 structures (51.57%), while the second cluster (Figure 14B) comprises 4843 structures (48.43%). This distribution indicates that the aromatic ring does not undergo full rotation around the C-2-C-3 bond, a result consistent with the ^1^H NMR spectrum.

For the non-aromatic ring, six distinct groups of conformations were identified (Figure 15). Cluster 1 comprises 10 structures (0.10%) in a boat conformation (Figure 15A). The average angle between each carbon–hydrogen bond and the plane that minimises the distance to all the carbon atoms in the ring shows that C-1′-H-1, C-2′-H-1, C-3′-H-1, C-4′-H-2, C-5′-H-1, and C-6′-H-1 are in axial positions (−68.63° ± 7.03°, 62.61° ± 12.56°, 62.74° ± 10.97°, −74.62° ± 5.67°, 70.36° ± 18.11°, and 57.86° ± 16.94°, respectively), while C-2′-H-2, C-3′-H-2, C-4′-H-1, and C-5′-H-2 are in equatorial positions (−22.76° ± 21.26°, −28.68° ± 21.99°, 24.87° ± 8.94°, and −28.91° ± 20.21°, respectively). Cluster 2 is the dominant group, consisting of 8726 structures (87.25%) in a chair conformation (Figure 15B). The carbon–hydrogen mean bond angles relative to the ring plane indicate that C-1′-H-1, C-2′-H-1, C-3′-H-2, C-4′-H-1, C-5′-H-2, and C-6′-H-1 are in axial positions (−76.98° ± 4.46°, 79.69° ± 4.3°, −74.84° ± 6.19°, 79.30° ± 4.38°, −76.35° ± 4.28°, and 78.05° ± 5.44°, respectively), while C-2′-H-2, C-3′-H-1, C-4′-H-2, and C-5′-H-1 are in equatorial positions (−17.41° ± 6.68°, 30.36° ± 6.61°, −12.93° ± 6.54°, and 16.9° ± 6.62°, respectively). Cluster 3 contains 1134 structures (11.34%) in a chair conformation (Figure 15C). The carbon–hydrogen bond angles relative to the ring plane indicate that C-2′-H-2, C-3′-H-1, C-4′-H-2, and C-5′-H-1 are in axial positions (−74.81° ± 5.74°, 82.12° ± 4.49°, −75.68° ± 5.71°, and 77.56° ± 4.76°, respectively), while C-1′-H-1, C-2′-H-1, C-3′-H-2, C-4′-H-1, C-5′-H-2, and C-6′-H-1 are in equatorial positions (−35.92° ± 7.3°, 24.21° ± 6.98°, −16.97° ± 7.28°, 27.72° ± 6.57°, −15.53° ± 6.88°, and 14.57° ± 7.52°, respectively).

Cluster 4 consists of 33 structures (0.33%) in a boat conformation (Figure 15D). The carbon–hydrogen bond angles relative to the ring plane indicate that C-1′-H-1, C-2′-H-2, C-3′-H-1, C-4′-H-2, C-5′-H-2, and C-6′-H-1 are in axial positions (−76.51° ± 6.56°, −64.36° ± 4.65°, 72.23° ± 6.97°, −52.05° ± 8.22°, −80.27° ± 6.3°, and 82.61° ± 3.96°, respectively), while C-2′-H-1, C-3′-H-2, C-4′-H-1, and C-5′-H-1 are in equatorial positions (25.96° ± 5.93°, −0.43° ± 7.62°, 42.12° ± 7.39°, and 18.09° ± 10.7°, respectively). Cluster 5 includes 74 structures (0.74%) in a boat conformation (Figure 15E). The carbon–hydrogen bond angles relative to the ring plane indicate that C-1′-H-1, C-2′-H-2, C-3′-H-1, C-4′-H-1, C-5′-H-2, and C-6′-H-1 are in axial positions (−55.93° ± 9.3°, −75.61° ± 5.53°, 74.11° ± 6.53°, 63.65° ± 6.34°, −72.52° ± 6.19°, and 69.93° ± 6.65°, respectively), while C-2′-H-1, C-3′-H-2, C-4′-H-2, and C-5′-H-1 are in equatorial positions (15.78° ± 7.12°, −19.13° ± 10.83°, −25.86° ± 8.75°, and 3.33° ± 6.54°, respectively). Cluster 6 also contains 24 structures (0.24%) in a boat conformation (Figure 15F). The carbon–hydrogen bond angles relative to the ring plane indicate that C-1′-H-1, C-2′-H-1, C-3′-H-2, C-4′-H-2, and C-5′-H-1 are in axial positions (−63.55° ± 6.11°, 77.19° ± 4.93°, −70.0° ± 8.4°, −54.64° ± 13.45°, and 76.78° ± 7.16°, respectively), while C-2′-H-2, C-3′-H-1, C-4′-H-1, C-5′-H-2, and C-6′-H-1 are in equatorial positions (−6.36° ± 6.46°, 31.53° ± 8.9°, 50.58° ± 12.56°, −15.77° ± 9.52°, and 32.19° ± 6.31°, respectively).

## 3. Discussion

The opioid crisis of the XXI century, particularly involving non-fentanyl synthetic opioids such as U-type opioids and nitazenes, is an emerging problem that endangers public health. The fact that there is a cyclical phenomenon of new compounds being synthesised to circumvent legislation, followed by their respective prohibition, as well as the existence of structural isomers and the lack of standards, makes the identification and characterisation of these compounds challenging according to forensic guidelines. Here, we combined GC-MS, NMR, and MDS to address this issue and characterise a substance delivered to our laboratory by the SCANNER project for confirmation.

### 3.1. GC-MS

By comparing the mass spectra from the GC-MS analysis with the Chemical C. Cayman Spectral Library (v21022022) and SWGdrug (v3.9) databases, it was established that the compound under study was the U-type opioid U-51754 or its structural isomer, U-48800. These two compounds differ only in the position of the aromatic ring chlorine atoms. In U-51754, the chlorine atoms are located at the C-5 and C-6 positions, while in U-48800, the chlorine atoms are located at the C-4 and C-6 positions. Mass spectrum analysis confirmed that there were two chlorine atoms in the structure. However, the fragmentation pattern did not make it possible to distinguish between the two isomers. Consequently, this technique did not allow the molecule to be characterised unequivocally.

### 3.2. NMR

The sample was then analysed using NMR, namely, ^1^H, ^13^C APT, COSY, TOCSY, HSQC, HMBC, and NOESY sequences. Both U-48800 and U-51754 molecules contain a 1,2,4-trisubstituted benzene moiety. Also, both molecules originate from an ABX spin system assignment, which is present in the ^1^H NMR spectrum with three aromatic signals. Although both compounds have an ABX spin system, the HMBC correlation of the H_A_ signal at 7.37 ppm (*J* = 8.2 Hz) with the carbon at 39.55 ppm is only compatible with the structure of U-48000, allowing the NMR aromatic methine signals at ^1^H/^13^C 7.49/129.96, 7.33/128.28, and 7.37/134.20 ppm to be assigned to positions C-5, C-7, and C-8, respectively. Nevertheless, the spectra were fully assigned for future reference.

### 3.3. MDS

The molecular dynamics simulation was performed to obtain the 3D structure of U-48800 and its possible conformers in methanol to match NMR experimental conditions. The analysis allowed for clustering U-48800 into two different conformers based on the structure of the aromatic ring and into six different conformers based on the structure of the non-aromatic ring.

The molecular simulation allowed us to better understand the AB split pattern of the C-2 methylene group by evaluating the accessible rotations of the C1-C-2-C-3-C4 dihedral. Figure 14 shows the obtained structures in superimposition, making it evident that there is no full rotation between the C-1-C-2 and C-2-C-3 bonds. The strong bond rotation constraint observed in the molecular dynamics explains the AB split pattern of the ^1^H band at 4.03 and 3.96 ppm, which is inferred to arise from a lack of rotation between the C-1-C-2 and C-2-C-3 bonds.

In the non-aromatic ring, six different conformers were found in the molecular simulation. These conformers assume chair conformations (Figure 15B,C) and a boat conformation (Figure 15A,D–F). The most representative conformations, shown in Figure 15B,C, correspond to 83.82% and 11.34% of the structures sampled in the simulation. In these structures, the non-aromatic ring is in the chair conformation, with the carbons C-3′, C-6′, and C-1′, C-4′ positioned above the ring plane, respectively. This is not surprising since the chair conformation is the most stable conformation for cyclohexane. In one conformer, the C-1′-H-1, C-2′-H-1, C-3′-H-2, C-4′-H-1, C-5′-H-2, and C-6′-H-1 protons are in axial positions, and the C-2′-H-2, C-3′-H-2, C-4′-H-1, and C-5′-H-2 protons are in equatorial positions (Figure 15B). In the other conformer, the C-2′-H-2, C-3′-H-1, C-4′-H-2, and C-5′-H-1 protons are in axial positions, and the C-1′-H-1, C-2′-H-1, C-3′-H-2, C-4′-H-1, C-5′-H-2, and C-6′-H-1 protons are in equatorial positions relative to the ring.

The generated structures allow us to better perceive the angular positions of the CH_2_ protons in the cyclohexane ring, i.e., the axial or equatorial protons in this ring, which consequently confirms the NMR assignment of this part of the molecule.

### 3.4. Final Considerations: Key Steps to Identify U-Type Opioids Without Standards

Identifying and confirming a substance according to the forensic guidelines requires more than one analytical technique and standards. These prerequisites involve the unequivocal identification of the substance, avoiding reasonable doubt. NPS poses considerable challenges to these working procedures due to the unavailability of standards in the laboratory and the novelty of the molecules. In this case, and according to ENFSI, when comparative data are not available, such as for new NPS or isomers of existing controlled substances, “it is up to the individual laboratory/drug analyst to determine which additional analytical technique(s) may be necessary to identify the substance in question” [9] (p. 8). Thus, the procedure is vague and not standardised. Moreover, some of the data available on the internet are in the form of non-peer-reviewed reports showing spectral data without any interpretation [21,22,23,24,25]. To the best of our knowledge, the identification of U-type opioids in the literature is performed using hyphenated mass spectrometry techniques (GC-MS and LC-MS), NMR, and single-crystal X-ray diffraction [11,12,13,14,15,16,17]. Most of the forensic science laboratories worldwide do not have access to all these techniques. In addition, forensic science responses must be accurate and rapid simultaneously, meaning that the procedures must be well-defined using the least number of techniques possible. The present study proposes a workflow to unequivocally identify U-type opioids without standards. It is worth mentioning that without standards, structural characterisation of NPS is mandatory since legislation often applies to only one of the isomers.

## 4. Materials and Methods

### 4.1. Drug Sample

The suspect sample (a white powder) was submitted for analysis by Kosmicare’s drug-checking laboratory as part of an interlaboratory test promoted by the EU-funded project SCANNER (Grant agreement No: 861834). 

### 4.2. Reagents and Chemicals

For GC-MS analysis, HPLC-grade methanol (MeOH, >99.9%) was purchased from Panreac (Panreac, Barcelona, Spain). For NMR analysis, deuterated methanol (MeOD) was purchased from Sigma-Aldrich (Sigma-Aldrich, St. Louis, MO, USA).

### 4.3. Sample Preparation

For GC-MS analysis, 2.5 mg of the sample was dissolved in 1 mL of MeOH and vortexed for 1 min. The extract was then centrifuged to remove insoluble material and directly injected into the GC-MS system. Additionally, 10 mg of the sample was dissolved in 0.6 mL of MeOD to obtain NMR spectra. 

### 4.4. Instrumentation

The analysis by GC-MS was performed on an Agilent 6890 series gas chromatography system, coupled with a 5973 series mass selective detector and a 7683 series automated injector (Agilent Technologies, Santa Clara, CA, USA). The GC separation was performed on a Mega 5MS column (30 m × 0.25 mm × 0.25 μm). The oven temperature programme started at 90 °C and was held for 2 min, and then it was increased by 20 °C/min until it reached 315 °C, which was maintained for 5 min. Helium was used as the carrier gas at a flow rate of 0.7 mL/min. A total of 2 μL of the previously prepared sample was injected in split mode (50:1) at 280 °C. The transfer line temperature was set to 280 °C, the ion source temperature was set to 230 °C, and the quadrupole temperature was set to 150 °C. The mass spectrometer electron ionisation was set to 70 eV, and the run was performed in scan mode over the *m*/*z* range of 30–450. The compound’s mass spectrum was identified using the Chemical C. Cayman Spectral Library (v21022022) and the SWGdrug database (v3.9). 

The analysis by NMR was performed using a Brucker Avance II 400 MHz Spectrometer (Bruker BioSpin GmbH, Rheinstetten, Germany). The sample solution was transferred to a 5 mm NMR tube, and the NMR spectra were acquired, operating at 400.13 MHz for ^1^H NMR and 100.61 MHz for ^13^C APT NMR. Chemical shifts (δ) were expressed as parts per million (ppm) and referenced to the signal of MeOD (δ_H = 3.3 ppm, δ_C = 49.0 ppm). Coupling constants (*J*) were reported in Hertz (Hz) units. Structure elucidation through the respective assignment of the carbon and proton signals was based on the analysis of NMR spectra obtained using 1D (^1^H and ^13^C APT) and 2D ([^1^H-^1^H]-COSY, [^1^H-^1^H]-TOCSY, [^1^H-^13^C]-HSQC, and [^13^C-^1^H]-HMBC) techniques. 

### 4.5. Molecular Dynamic Simulation 

The isomeric Simplified Molecular Input Line Entry System (SMILES) representation of the synthetic opioid U-48800 (CID: 137700072) was retrieved from the PubChem database (using its PubChem Compound Identifier (CID) with the PubChemPy v.1.0.4 library for Python v.3.12.3) [26,27]. Subsequently, the initial three-dimensional molecular structure was constructed by adding all hydrogen atoms to the molecule using the Open Babel v.3.1.1 bindings for Python [28]. Using this initial structure, we applied the systematic rotor search method from Open Babel with the “mmff94” force field and 1000 steps to identify a low-energy conformer, which served as a starting point for the molecular dynamics simulation.

LigParGen was used to generate the topology (.top), force field (.itp), and structure (.gro) files for the U-48800 provided structure based on the OPLS-AA force field [29,30,31]. Subsequently, a molecular dynamics simulation of the replica exchange with solute tempering protocol (REST2) was carried out using Gromacs v.2023.5 patched with plumed v.2.9.1 [32,33]. This protocol is a variation of replica exchange molecular dynamics (REMD) that selectively scales interactions involving the solute, enhancing the sampling of solute conformations while minimising the computational cost. The simulation system was prepared by creating a cubic box with 1 nm between the solute and the box edge. Seven hundred twenty-eight molecules of methanol surrounded the solute. The system energy was minimised for 10.000 steps. Subsequently, the system was equilibrated in the NVT ensemble (constant number of particles, volume, and temperature) for 100 ps and then in the NPT ensemble (constant number of particles, pressure, and temperature) for another 100 ps. The final simulation was run in the NPT ensemble for 10 ns, using 10 replicas, with replica exchange attempts every 1 ps, and lambda values of 1.0, 0.928, 0.862, 0.8, 0.742, 0.689, 0.64, 0.594, 0.551, and 0.511, calculated from the following geometric progression of temperatures: 293.15 K, 315.82 K, 340.25 K, 366.56 K, 394.91 K, 425.46 K, 458.36 K, 493.81 K, 532.0 K, and 573.15 K, respectively, as previously described by Sunhwan Jo and Wei Jiang [34]. This allowed us to obtain an exchange rate between replicas that ranged from 0.72 to 0.77. The parameters used for minimisation, equilibration, and the production run can be found in more detail in Appendix B.

Following the simulation, the trajectories of all replicas were concatenated, and the solute was centred in the simulation box. The resultant trajectory file was subsequently used to analyse the rotation of the aromatic ring about the bond C-2-C-3 and the conformations adopted by the non-aromatic ring throughout the simulation.

To evaluate the rotation of the aromatic ring about the C-2-C-3 bond, PyMol v.3.0.0 was used to compute the dihedral angle involving C-1-C-2-C-3-C-4 for each conformer present in the trajectory [35]. Based on the calculated angles, structures were clustered using a 0-degree angle as a cutoff point (a value chosen based on the density plot of the angles).

All conformations in the trajectory file were aligned to the first state, based on the carbon atoms that constitute the ring, to evaluate the non-aromatic ring conformations. Subsequently, the root mean squared deviation between structures was computed using all the ring carbons and the atoms connected to these carbons. The resulting distance matrix was used to cluster the structures into an initial set of 14 groups using the agglomerative clustering function provided by Sklearn v.1.13.0 [36]. These clusters were subsequently analysed visually using PyMol and, whenever applicable, merged, which resulted in a final set of 6 groups. To evaluate the orientation of the hydrogens in relation to the non-aromatic ring, the Numpy v.1.26.4 library, in conjunction with Python’s math and statistics libraries, was used to compute angles formed between the plane that minimised the distance to all the carbon atoms of the ring and the line formed by each of these carbons and the corresponding bounded hydrogens [37].

Images of each cluster were generated using PyMol after the superimposition of all conformers, which were previously aligned based on the carbons of the aromatic ring (Figure 14) or the non-aromatic ring (Figure 15).

Jupyter notebooks running on a Jupyter server v.2.14.0 containing the three-dimensional structure preparation, molecular dynamics setup, and subsequent analysis can be found at https://github.com/carlosfamilia/MDS-U48800 (accessed on 10 December 2024).

## 5. Conclusions

Forensic laboratories frequently receive seized samples without prior identification, making the detection of new psychoactive substances (NPS) particularly challenging. Relying solely on colorimetric tests poses a high risk of false positive or negative results [38]. A more powerful method of putative identification is usually needed for such cases, such as GC-MS or LC-MS. These techniques allow for presumptive identification based on databases. Furthermore, fragment pattern analysis allows us to characterise the presence and quantity of halides, among other molecular features [39]. In this particular case, the molecule is not in the database. The chromatographic step provides information on the presence or absence of organic contamination. In the absence of contaminants, the compound proceeds directly to NMR spectroscopy. Otherwise, the sample must be purified.

The present study showed the importance of combining MS, NMR, and bioinformatics tools to identify position isomers of utopioids in forensic laboratories without reference materials. The techniques employed allowed for the unequivocal structural identification of isomer U-48800. This methodology was developed to address a real drug-checking case that has involved different laboratories across different countries. The same approach can be applied to distinguish other types of controlled from uncontrolled structural isomers, a crucial issue due to the various consequences arising from their legal framework.

## Figures and Tables

**Figure 1 ijms-26-02219-f001:**
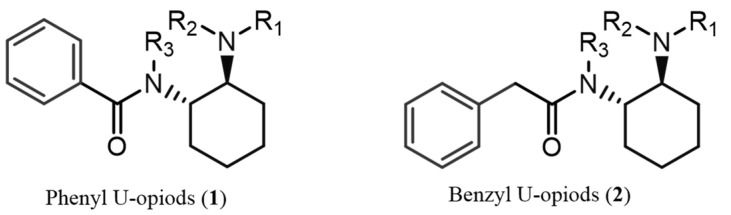
Generic structures of U-type opioid compounds: (**1**) phenyl U-opioids, (**2**) benzyl U-opioids. These utopioid scaffolds can appear with several types of substitutions: (i) the aromatic ring by 1 or 2 halogen atoms or a trifluoromethyl or methylenedioxy group; (ii) R_3_ = H or an alkyl (e.g., methyl, ethyl, or isopropyl); (iii) R_1_ and R_2_ are usually the same alkyl substituent (e.g., methyl, ethyl, or belonging to a pyrrolidinyl group); (iv) the cyclohexyl by tetrahydrofuran (THF), forming a spirocyclic moiety.

**Figure 2 ijms-26-02219-f002:**
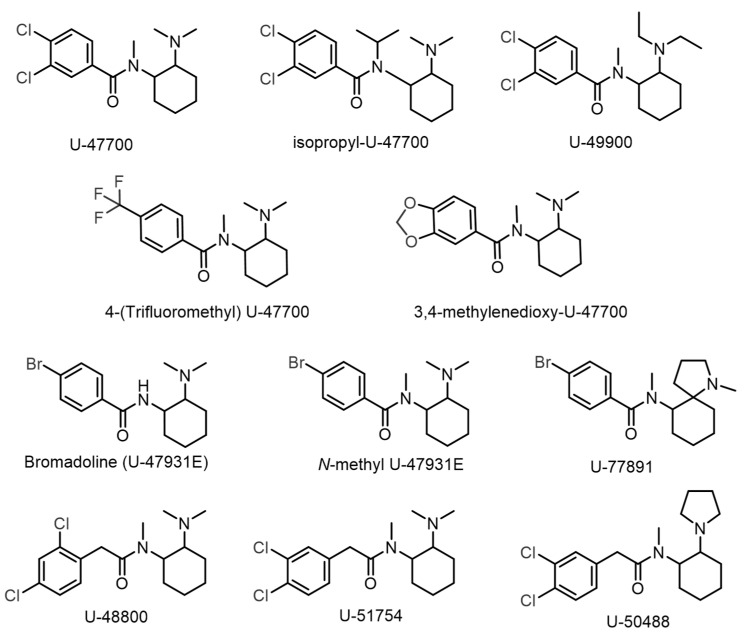
U-type opioids currently found in illicit drug markets.

**Figure 3 ijms-26-02219-f003:**
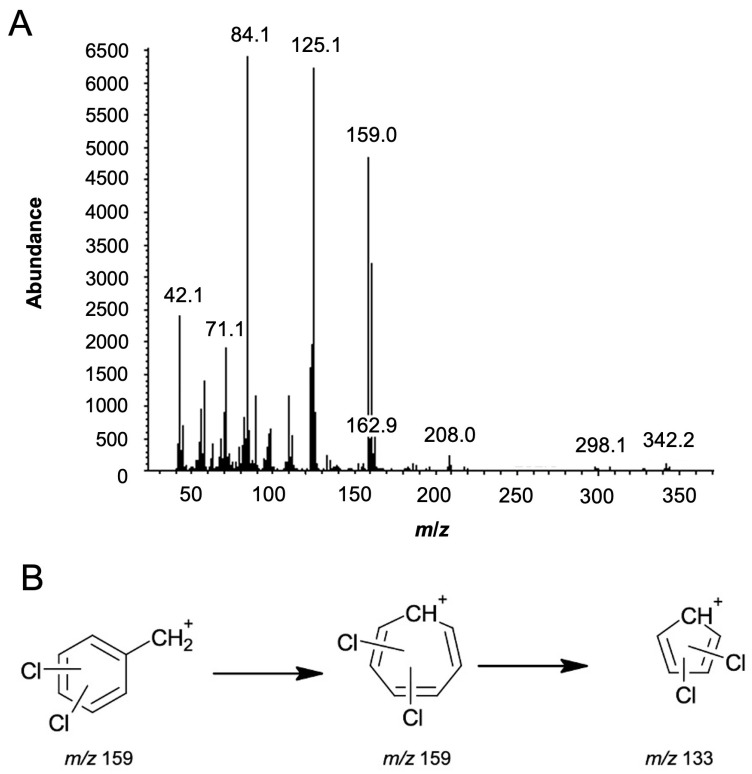
(**A**) Mass spectrum of utopioid sample analysed using GC-MS. (**B**) Possible *m/z* 159 fragment and its transformation into the *m/z* 133 fragment.

**Figure 4 ijms-26-02219-f004:**
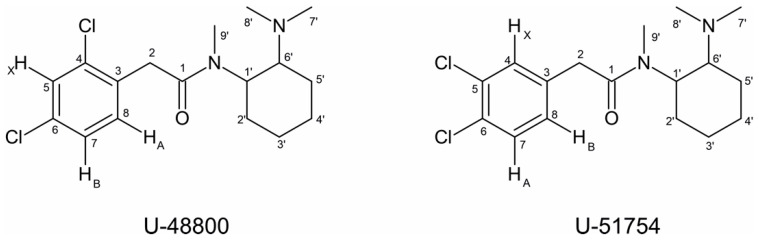
Molecular structures of U-48800 and U-51754 with all carbons numbered.

**Figure 5 ijms-26-02219-f005:**
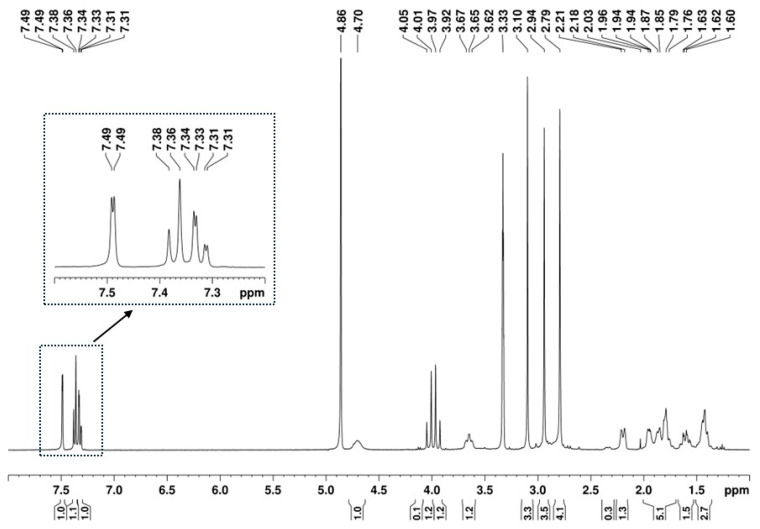
^1^H NMR (400 MHz, MeOD) spectrum of the utopioid sample.

**Figure 6 ijms-26-02219-f006:**
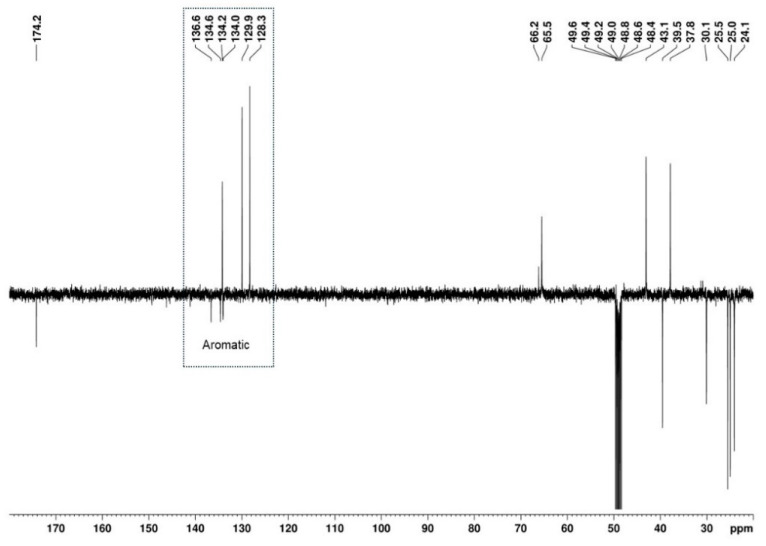
^13^C APT (100 MHz, MeOD) spectrum of the utopioid sample.

**Figure 7 ijms-26-02219-f007:**
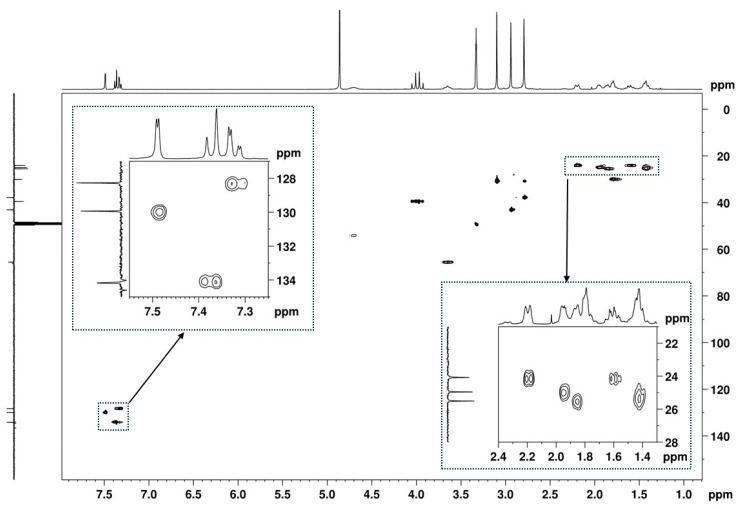
[^1^H-^13^C]-HSQC spectrum (MeOD) of the utopioid sample with two zooming zones.

**Figure 8 ijms-26-02219-f008:**
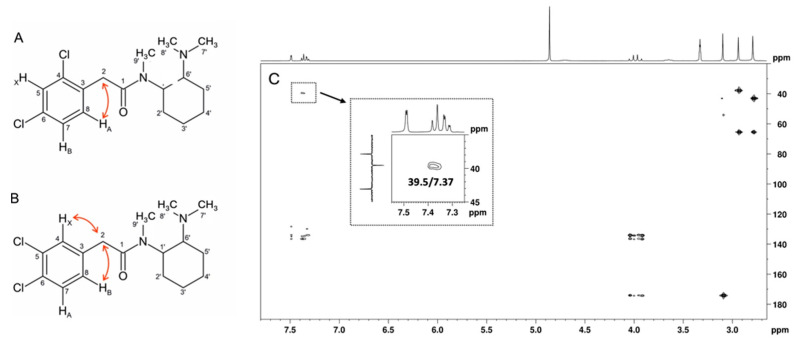
(**A**) Structure of U-48800 and (**B**) structure of U-51754 with key [^13^C-^1^H]-HMBC correlations of each ABX spin system and (**C**) HMBC full spectrum of the utopioid sample with C-2/H_A_ (δ_13C_ 39.5/δ_1H_ 7.37) correlation zooming.

**Figure 9 ijms-26-02219-f009:**
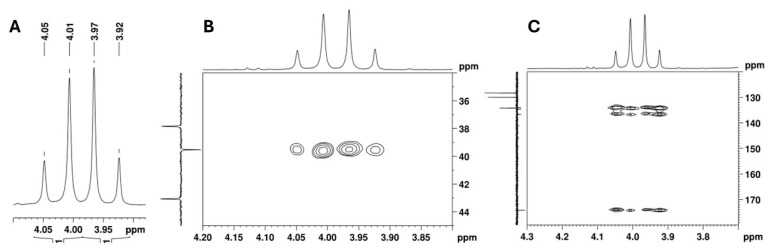
NMR spectra expansion of the utopioid sample at the region of the methylene AB spin system: (**A**) ^1^H NMR, (**B**) [^1^H-^13^C]-HSQC, and (**C**) [^13^C-^1^H]-HMBC.

**Figure 10 ijms-26-02219-f010:**
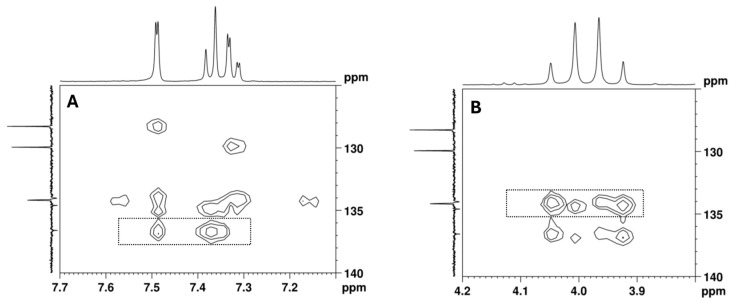
[^13^C-^1^H]-HMBC spectrum of the utopioid sample expanded in the regions of the key correlations of the quaternary aromatic carbons with (**A**) aromatic protons and (**B**) C-2 methylene protons.

**Figure 11 ijms-26-02219-f011:**
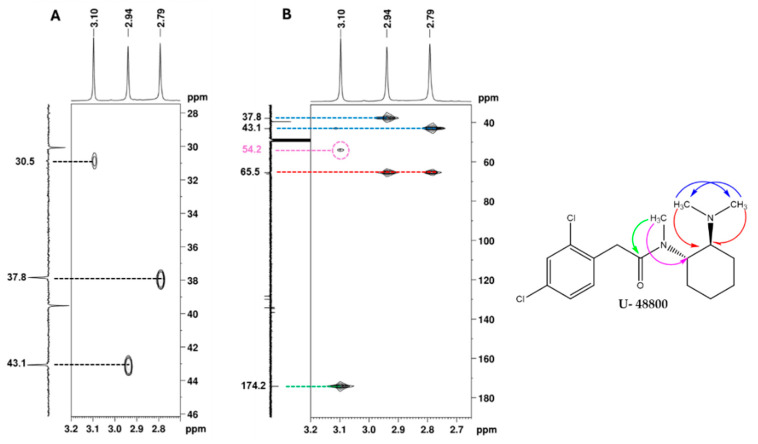
NMR spectra of the utopioid sample expanded in the regions of the key correlations of the *N*-methyl groups: (**A**) [^1^H-^13^C]-HSQC, (**B**) [^13^C-^1^H]-HMBC and structure of U-48800 showing key [^13^C-^1^H]-HMBC correlations. In the figure, blue arrows show ^13^C-^1^H correlations between 7′ and 8′ methyl groups; red arrows show ^13^C-^1^H correlations between 7′ and 8′ methyl groups with 6′ methine group; the green arrow shows ^13^C-^1^H correlations between 9′ methyl group and 1 carbonyl group, and the pink arrow shows ^13^C-^1^H correlations between 9′ methyl group and 1′ methine group.

**Figure 12 ijms-26-02219-f012:**
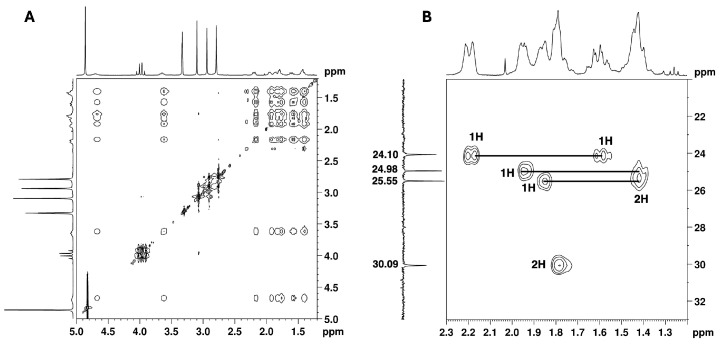
NMR spectra of the utopioid sample expanded in the regions of cyclohexane moiety: (**A**) [^1^H-^1^H]-TOCSY and (**B**) [^1^H-^13^C]-HSQC of the methylene protons.

**Figure 13 ijms-26-02219-f013:**
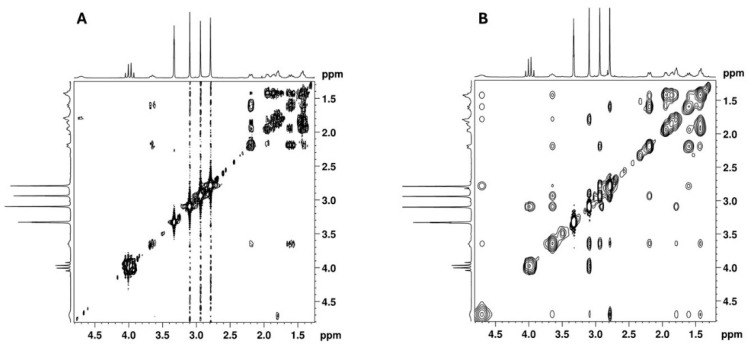
(**A**) [^1^H-^1^H]-COSY and (**B**) [^1^H-^1^H]-NOESY spectra of cyclohexane moiety of the utopioid sample expanded in the zone of cyclohexane moiety.

**Figure 14 ijms-26-02219-f014:**
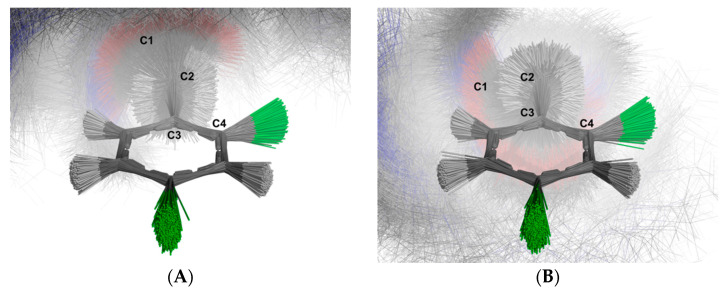
Schematic depiction illustrating the superimposition of the U-48800 conformers extracted from the trajectory obtained through molecular dynamics simulations using methanol as solvent. The alignment of all structures was carried out using the carbon atoms of the aromatic ring. The labelled clusters represent distinct conformations based on the C-2 rotation: (**A**) Conformations with negative angles, (**B**) Conformations with positive angles. Carbon atoms are shown in dark grey, while hydrogen atoms are represented in white. Oxygen atoms are depicted in red, chloride atoms are shown in green and nitrogen atoms in blue.

**Figure 15 ijms-26-02219-f015:**
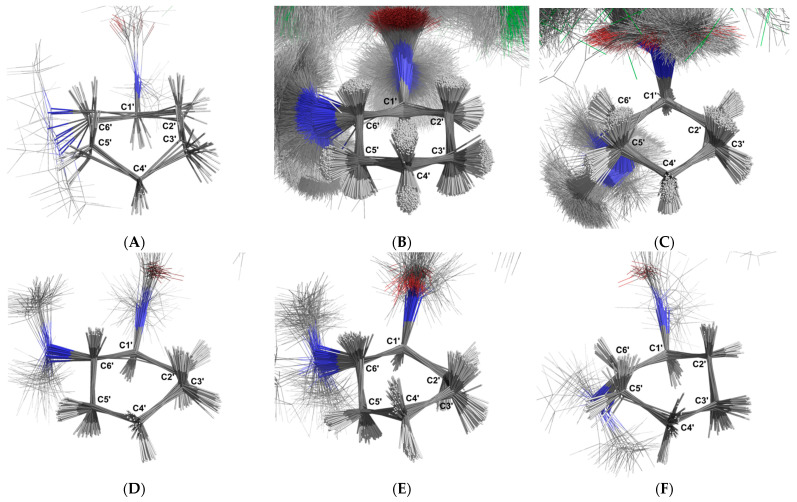
Schematic depiction illustrating the superimposition of the U-48800 conformers extracted from the trajectory obtained through molecular dynamics simulations using methanol as a solvent. The alignment of all structures was carried out using the carbon atoms of the cyclohexane group. The labelled clusters represent distinct conformations adopted by the cyclohexane molecule: (**A**) Boat conformation with C-1′ and C-4′ below the ring plane, (**B**) Chair conformation with C-3′ below and C-6′ above the ring plane, (**C**) Chair conformation with C-1′ above and C-4′ below the ring plane, (**D**) Boat conformation with C-3′ and C-6′ above the ring plane, (**E**) Boat conformation with C-2′ and C-5′ below the ring plane, and (**F**) Boat conformation with C-2′ and C-5′ above the ring plane. Carbon atoms are shown in dark grey, while hydrogen atoms are represented in white. Oxygen atoms are depicted in red and nitrogen atoms in blue.

**Table 1 ijms-26-02219-t001:** NMR data of U-48800 in MeOD ^a^.

Position ^b^	δ ^13^C, Type ^c^	δ ^1^H, Integration, Multiplicity, *J* (Hz) ^d^	[^1^H-^1^H]-COSY ^e^	[^13^C-^1^H]-HMBC ^e^
1	174.2, C	-	-	4.03, 3.94, 3.09
2	39.5, CH_2_	4.03, 1H, d (16.7), H_a_ 3.94, 1H, d (16.7), H_b_	-	7.37
3	134.6 *, C	-	-	7.49, 7.37, 4.03, 3.09
4	136.6, C	-	-	7.49, 7.37, 4.03, 3.09
5	129.9, CH	7.49, 1H, d (1.9)	7.32	7.32
6	134.0 *, C	-	-	7.49, 7.32
7	128.3, CH	7.32, 1H, dd (8.2, 1.9)	7.37, 7.49	7.49
8	134.2, CH	7.37, 1H, d (8.3)	7.32	7.32, 4.03, 3.09
1′	54.2, CH	4.70, 1H, broad s, H_ax_	3.65, 1.79	3.09
2′	30.1, CH_2_	1.79, 2H	4.70, 1.42	-
3′	25.5, CH_2_	1.86, 1H, m, H_eq_ 1.42, 2H *, m, H_ax_	1.95, 1.42 2.19, 1.95, 1.86, 1.79,1.61	- -
4′	25.0, CH_2_	1.95, 1H, m, H_eq_ 1.42; 2H *, m, H_ax_	2.19, 1.86, 1.61, 1.42 2.19, 1.95, 1.86, 1.79, 1.61	- -
5′	24.1, CH_2_	2.19, 1H, H_eq_ 1.61, 1H, H_ax_	3.65, 1.95, 1.61, 1.42 3.65, 2.19, 1.95, 1.42	- -
6′	65.5, CH	3.65; 1H; m, H_ax_	4.70, 2.19, 1.61	2.94, 2.79
7′/8′	43.1, CH_3_ 37.8, CH_3_	2.94, 3H, s 2.79, 3H, s	-	2.79 2.94
9′	30.9; CH_3_	3.09; 3H; s	-	-

^a^ Bruker AVANCE 400 spectrometer, chemical shifts (δ, ppm) referred to as MeOD (3.30 for ^1^H and 49.0 for ^13^C). ^b^ Assignments are supported by 2D NMR experiments: COSY, HSQC, HMBC, and NOESY. ^c^ Type established by ^13^C APT. ^d^ Connectivity established by HSQC. (multiplicity: s-singlet, d-duplet, dd-double duplet, t-triplet, q-quartet, m-multiplet), *J* (coupling constant, Hz). ^e^ Only key correlations are presented. * The attribution may be reversed.

## Data Availability

Data are available from the corresponding author upon request.

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
