# Peer review of "Drug-Checking and Monitoring New Psychoactive Substances: Identification of the U-48800 Synthetic Opioid Using Mass Spectrometry, Nuclear Magnetic Resonance Spectroscopy, and Bioinformatic Tools"

_ijms, 2025, doi:10.3390/ijms26052219_

Round 1
Reviewer 1 Report
Comments and Suggestions for Authors
In this study, the authors' team used a combination of gas chromatography-mass spectrometry (GC-MS), nuclear magnetic resonance spectroscopy (NMR), and molecular dynamics to definitively characterize U-type opioid samples (suspected to contain U-48800 or U-51754) received by Kosmicare from the EU-funded SCANNER project, and definitively detected the presence of U-48800 in the samples, providing a solution for the identification of this psychoactive substance. This article is acceptable if the following suggestions can be adopted.
1. There seems to be some problems with the display of the annotation text format in Table 1, and I hope it can be corrected.
2. Personal suggestions can be put into "4. Materials and Methods" in "2. Results", which is more in line with the logic of reading.
3. What is the cost of the assay used in this study?
4. Are there any plans for further process simplification in the future?
5. Would you be willing to replicate the methods in this study to other labs? What support is needed from which departments?
Reviewer 2 Report
Comments and Suggestions for Authors
In this manuscript, the author revealed the identification of U-48800 Synthetic opioids via MS, 3 NMR, and Bioinformatic Tools, underscoring the importance of a structural characterization strategy in forensic laboratories. I believe the manuscript is suitable for publication after some minor issues are addressed.
comment;
1. Figure image clarity is not clear and highlights that particular value with high resolution.
2. In figures author should mention which solvent they used for analysis for prediction.
3. The NMR window should be 0-10 ppm in general all cases.
Comments on the Quality of English LanguageEnglish should be rechecked in some paragraphs
Round 2
Reviewer 1 Report
Comments and Suggestions for Authors
Congratulations on the successful publication of your paper. I have no further questions. Great work!
Comments on the Quality of English Languageacceptable